# Immune Escape Mechanisms and Their Clinical Relevance in Head and Neck Squamous Cell Carcinoma

**DOI:** 10.3390/ijms21197032

**Published:** 2020-09-24

**Authors:** Barbara Seliger, Chiara Massa, Bo Yang, Daniel Bethmann, Matthias Kappler, Alexander Walter Eckert, Claudia Wickenhauser

**Affiliations:** 1Institute of Medical Immunology, Martin Luther University Halle-Wittenberg, 06112 Halle, Germany; chiara.massa@uk-halle.de (C.M.); bo.yang@uk-halle.de (B.Y.); 2Fraunhofer Institute of Cell Therapy and Immunology, 04103 Leipzig, Germany; 3Institute of Pathology, Martin Luther University Halle-Wittenberg, 06112 Halle, Germany; daniel.bethmann@uk-halle.de (D.B.); claudia.wickenhauser@uk-halle.de (C.W.); 4Department of Oral and Maxillofacial Plastic Surgery, Martin Luther University Halle-Wittenberg, 06120 Halle (Saale), Germany; matthias.kappler@uk-halle.de (M.K.); alexander.eckert@uk-halle.de (A.W.E.); 5Klinik für Mund-, Kiefer- und Plastische Gesichtschirurgie, Universitätsklinik der Paracelsus Medizinischen Privatuniversität; 90471 Nürnberg, Germany

**Keywords:** head and neck squamous cell carcinoma, immune escape, tumor microenvironment, immune responses, immunotherapy

## Abstract

Immunotherapy has been recently approved for the treatment of relapsed and metastatic human papilloma virus (HPV) positive and negative head and neck squamous cell carcinoma (HNSCC). However, the response of patients is limited and the overall survival remains short with a low rate of long-term survivors. There exists growing evidence that complex and partially redundant immune escape mechanisms play an important role for the low efficacy of immunotherapies in this disease. These are caused by diverse complex processes characterized by (i) changes in the expression of immune modulatory molecules in tumor cells, (ii) alterations in the frequency, composition and clonal expansion of immune cell subpopulations in the tumor microenvironment and peripheral blood leading to reduced innate and adaptive immune responses, (iii) impaired homing of immune cells to the tumor site as well as (iv) the presence of immune suppressive soluble and physical factors in the tumor microenvironment. We here summarize the major immune escape strategies of HNSCC lesions, highlight pathways, and molecular targets that help to attenuate HNSCC-induced immune tolerance, affect the selection and success of immunotherapeutic approaches to overcome resistance to immunotherapy by targeting immune escape mechanisms and thus improve the HNSCC patients’ outcome.

## 1. Introduction

Head and neck squamous cell carcinoma (HNSCC) represents the sixth most common cancer worldwide, with 700,000 new cases diagnosed in 2019 with a mortality rate ranging between 40–50% [1,2]. It comprises a heterogeneous epithelial neoplasia of the oral cavity, oro-, naso- and hypopharynx, nasal cavity and larynx [3]. The development of HNSCC has been correlated to life-style risk factors like smoking and alcohol consume as well as to infections with high risk human papilloma viruses (HPV) [4,5]. It is well accepted that HNSCC should be divided into HPV-negative and HPV-positive diseases, which differ regarding their prevalence, their molecular signature, localization, prognosis, therapy response and immune cell infiltration [6,7,8,9] (Table 1). Conventional treatment for HNSCC comprises of surgery, radio- and/or chemotherapy, which is associated with a substantial morbidity and toxicity rate. Therefore, more effective therapies are urgently needed. Recently, a number of targeted therapies using either monoclonal antibodies or tyrosine kinase inhibitors directed against the epidermal growth factor receptor (EGF-R), the vascular endothelial growth factor (VEGF) and its receptor (VEGF-R), the phosphatidyl-inositol-3-kinase (PI3K) and the mammalian target of rapamycin (mTOR) has been approved for the treatment of HNSCC, such as cetuximab, panitumumab, zalutumumab, gefitinib, erlotinib, lenvatinib, lapatinib, bevacizumab, sorafenib, rapamycin and everolimus, with significant, but still limited response rates ranging between 10–15%. The low efficacy to these targeted therapies might depend on the molecular and immunological make up as well as the heterogeneity of HNSCC lesions with a number of genetic drivers of oncogenesis altering also the anti-tumoral immune responses. During the last years, immunotherapy approaches have been shown to improve the survival of HNSCC patients, which include immunomodulatory antibodies, vaccines, oncolytic viruses and adoptive cell therapy. Treatment with immune checkpoint inhibitors (iCPi) targeting the programmed death receptor 1 (PD-1), like nivolumab and pembrolizumab and its ligand PD-L1 (atezolizumab), had significant clinical effects in a subset of HNSCC patients. However, the majority of HNSCC patients are resistant to these immunotherapies and the underlying molecular mechanisms of these resistances are so far mostly unknown [10,11,12]. It is currently discussed whether cellular and soluble factors of the tumor microenvironment (TME), alterations of the immunogenicity of tumor cells including cancer stem cells (CSC) per se, the composition of peripheral blood mononuclear cells (PBMNC) and/or the presence of immune suppressive mediators might lead to immune evasion.

Approximately 50 years ago, research suggested that pre-malignant cells frequently occur, but are spontaneously eliminated by the immune system before an invasive tumor can develop—thereby preventing neoplastic transformation. This hypothesis was supported by epidemiological data demonstrating that patients infected with the human immunodeficiency virus (HIV) and patients receiving immune suppressive therapy were more prone to cancer [13,14]. Furthermore, cancer patients can develop spontaneous tumor regressions [15], while an increased HNSCC incidence was detected in transplanted as well as immune suppressed patients [16]. These data led to the assumption that tumors can evade immune responses by developing so called immune escape strategies, which will allow the selection for and escape of non-immunogenic cells from immune surveillance by inhibiting the cytotoxic functions of immune cells via distinct mechanisms [17]. Thus, there exists an interplay between the immune system and cancer tissue, which was defined as cancer immunoediting [18,19,20,21]. Next to the direct immune evasion strategies of tumors, an immune suppressive network within the TME and changes in the immune cell repertoire and activity of peripheral blood were described. An increased understanding of these factors represents the basis and rationale for the development and design of novel individualized immunotherapies for cancer patients in the future [22]. In this review, the diverse processes leading to tumor immune escape in HNSCC and their impact on immunotherapies and combination strategies will be discussed.

## 2. Immune Escape Mechanisms of Tumors

It is generally accepted that an effective T cell activation is required for a proper anti-tumor response [23], which consists of two distinct signals. The first signal is mediated by the interaction of the T cell receptor (TCR) with tumor antigens presented by human leukocyte antigens (HLA) class I molecules, the second by the interaction of co-stimulatory receptors with their ligands resulting in T cell proliferation, cytokine secretion and cytotoxicity. A pre-existing intra-tumoral anti-tumor immune response has been associated with a favorable outcome and responsiveness to immunotherapies [24].

Tumor antigens can be classified into tumor-associated antigens (TAAs) or tumor-specific antigens (TSA) [25,26]. TAAs are expressed by both malignant and healthy cells, while TSA are expressed only by tumor cells. TSA and TAA are proteins, from which T cell epitopes are generated by the multi-catalytic proteasome complex consisting of the constitutive subunits MB1 (X), Delta (Y) and β2 (Z) as well as the interferon (IFN) inducible low molecular weight proteins (LMP) 2, 7 and 10. The yielded peptides are transported via the transporter associated with antigen processing (TAP) subunits TAP1 and TAP2 from the cytosol into the endoplasmic reticulum (ER) and then loaded onto HLA class I molecules. The trimeric complex consisting of the MHC class I heavy chain (HC), β_2_-microglobulin (β_2_-m) and the respective TAA/TSA derived epitope is then shuttled via the trans-Golgi to the cell surface and presented to CD8^+^ cytotoxic T lymphocytes (CTL) [27,28]. Complete loss or downregulation of HLA class I surface expression leads to evasion of tumor cells from CTL recognition. The loss or downregulation of HLA class I expression is often mediated by an abnormal expression and function of major components of the HLA class I antigen processing machinery (APM). This can be either due to structural abnormalities or deregulation at the transcriptional, epigenetic, post-transcriptional as well as post-translational level of APM components. In addition, the loss of adhesion molecules, e.g., ICAM-1, and the frequency of tumor infiltrating lymphocytes (TILs) interferes with anti-tumor responses [29,30]. Furthermore, non-classical HLA class I antigens, like HLA-G and HLA-E, may be expressed in tumors at a high frequency leading to tolerance not only against CTL, but also against natural killer (NK) cell-mediated cytotoxicity. In addition, immune suppressive ligands, like PD-L1 and nectin 4, and apoptosis-inducing ligands, such as the fas ligand and TRAIL, have been reported to be expressed by tumor cells and/or host’s myeloid cells, which results in apoptosis or anergy of T and/or NK cells.

The composition and frequency of immune cells within the TME and peripheral blood plays an important role in tumorigenesis. While NK cells, CD4^+^ and CD8^+^ T cells, dendritic cells (DC) and pro-inflammatory M1 macrophages promote anti-tumor immune responses, heterogeneous populations of myeloid-derived suppressor cells (MDSC), FOXP3^+^ regulatory T cells and M2 macrophages counteract tumor immunity [23,31]. The complexity of the immune system is reflected by the homing of immune cells, the composition of the immune cell infiltrate, the amount of inflammation and frequency of TILs, which affect the overall survival (OS) of patients and the efficacy of (immuno)therapy. These observations resulted in the categorization of inflamed, immune desert and immune excluded tumor phenotypes [32]. Tumors may also secrete various immune suppressive and anti-apoptotic factors like TGF-β, IL-10, PGE2 and IL-6 as well as release immune suppressive extracellular vesicles (EV), such as micro-vesicles and exosomes and different metabolites including arginase, adenosine, indolamine-2,3-dioxygenase (IDO) and nitric oxygen species (NOS) [33,34]. Furthermore, the immune system is also affected by metabolic changes in the TME due to hypoxic and acidic conditions [35]. A summary of major tumor-immune escape mechanisms are provided in Table 2.

The abovementioned strategies to escape immune surveillance, which are generally valid in malignant tumors, are also detected in HNSCC. Indeed, there exists evidence that HNSCC lesions exhibit a reduced immunogenicity as well as an enhanced immune dysfunction within the TME and in the peripheral blood of HNSCC patients [36]. In this context, both subgroups, HPV^−^ and HPV^+^ HNSCC, significantly differ regarding their molecular features, the degree of immune sup ression as well as their strategies to evade recognition and clearance by the host immune system. Furthermore, they differ in the qualitative and quantitative composition of the immune cell repertoire, of physical and soluble factors in the microenvironment as well as in the immune cell repertoire of peripheral blood [37], which are summarized in Table 3.

## 3. The Role of Classical and Non-Classical HLA Class I, Antigen Presentation and Processing and PD-L1 in HNSCC

The expression of classical HLA class I antigens was shown to be diminished or lost at a high frequency varying from 20% up to 70% of HNSCC lesions and cell lines, whereas β_2_-m expression was absent in approximately 16% of tumors [49,50,51,52]. 15–25% of primary HNSCC lesions and 40% of metastatic specimen had a total loss of HLA class I expression. There was no difference in the frequency of HLA class I abnormalities between HPV^-^ and HPV^+^ tumors [48]. Interestingly, the loss or downregulation of HLA class I and β_2_-m expression was correlated with the loss of chromosomal regions at 6p21.3 and a loss of heterozygosity (LOH) at 6p21.3 and 15q [51,53].

Reduced expression of HLA class I antigens was associated with a downregulation in the expression of various APM components. While TAP1, LMP2, LMP7 and β_2_-m expression were downregulated in up to 70% of HNSCC lesions, calnexin and ERp57 were only downregulated in approximately 25% of lesions [52,54,55]. The extent of HLA class I APM defects was associated with a low CD8^+^ T cell infiltration. It has also been demonstrated that overexpression and activation of epidermal growth factor receptor (EGF-R), which is expressed at a high frequency in HNSCC specimens, can lead to a downregulation of HLA class I by affecting downstream pathways, such as STAT1, MAPK and PI3K/AKT. This impaired HLA class I expression could in most publications be reversed by treatment with respective inhibitors or by IFN-γ [56]. Furthermore, a link between the expression of the chemokine CXCL14 and HLA class I expression was recently shown [57], demonstrating the ability of CXCL14 to restore the deficient HLA class I expression in HPV^+^ HNSCC cells. Regarding HLA class II expression, only a minority of HNSCC lesions, but a high frequency of stromal cells expressed HLA class II antigens with higher levels in HPV^+^ compared to HPV^-^ tumors [58].

Next to classical HLA class I antigens, the non-classical HLA-G expression was detected in more than 50% of HNSCC lesions, while healthy controls lacked HLA-G expression [51,52,59,60]. A higher frequency of HLA-G expression was found in HPV^+^ HNSCC when compared to HPV^-^ HNSCC [61]. Furthermore, significantly higher HLA-G expression levels were detected in metastases compared to primary tumors [62]. HLA-E expression was recently also found to be expressed on HNSCC [63]. Regarding PD-L1 expression, its frequency highly varied in the HNSCC tumors depending on the intra-tumoral localization and the tumor specification as PD-L1 expression was higher in the invasive tumor margin than in the tumor center and higher in HPV^+^ than in HPV^-^ HNSCC [48]. As for HLA-G, PD-L1 expression was more frequent in metastases than in primary tumors [64]. Interestingly, the metabolic tumor volume and the glycolytic activity correlated significantly positive with PD-L1 expression [65].

## 4. Tumor Microenvironment and HNSCC

A highly diverse spectrum of soluble and cellular factors within the TME of HNSSC have been described to be involved in the immune escape of this disease [66]. Thus, the TME has the ability to adapt to environmental demand depending on the tumor metabolism and hypoxia. This reprogramming not only determined the fate and functions of tumor cells, but also of immune cells. Both factors are known to play key roles in the course of HNSCC and to support the progression of this disease [67]. In reaction to hypoxia and acidic conditions, HNSCC switch to the glycolytic metabolism, which leads to the production of lactic acid and further reduces the pH [68,69]. Importantly, these metabolic alterations in the TME not only affect the HNSCC lesions by altering the expression of immune modulatory molecules, such as e.g., an increased expression of PD-L1, but also change the immune cell repertoire and activity. This leads to a reduced T cell activation, proliferation, cytotoxicity and diminished antigen processing of DC with simultaneous attraction and accumulation of immune suppressive cells like regulatory T cells (Tregs), MDSCs and tumor-associated macrophages (TAMs) [46,47]. This is accompanied by a high frequency of CD4^+^ Tregs, which was associated with an increased expression of tumor necrosis factor (TNF) receptor family members, but a reduced expression of genes involved in the IFN pathway. Interestingly, OX40, PD-1 and CTLA were enriched in T cells isolated from HNSCC [70]. MDSC were found to be recruited in HNSCC, but their role in this disease has not yet been analyzed in detail. Moreover, the hypoxic condition induces a stromal accumulation of TAMs, which are responsible for the *epithelial* mesenchymal *transition* (*EMT*) in this disease and might represent an immunotherapeutic target for HNSCC patients [71]. Furthermore, the hypoxia-induced release of VEGF and various chemokines could induce monocytes to differentiate into TAMs, while the secreted VEGF can also cause an abnormal angiogenesis [72,73]. This angiogenic switch further contributes to the maintenance of the hypoxic TME, additionally lowering the tissue pH.

Furthermore, a remodelling of the extracellular matrix (ECM) mediated by cancer-associated fibroblasts (CAFs) occurs, which leads to the protection of tumor cells by a biophysical barrier from effector T cells as well as to a resistance against monoclonal antibodies (mAbs) targeting EGF-R [74]. Interestingly, late stage HNSCC consists of more than 80% CAFs, which secrete hepatocyte growth factor thereby inducing a glycolytic switch in tumor lesions [75]. Furthermore, CAFs promote an immune suppressive TME through the induction of a pro-tumoral phenotype of macrophages [65]. This further reflects that different immune escape mechanisms are dependent of each other and crossover thereby creating a redundant and highly efficient immune suppressive system.

Recently, the immune landscape of HNSCC was analyzed using The Cancer Genome Atlas (TCGA) data and genetic profiles were identified, which delineate tumors in immune active and immune exhausted phenotypes [76]. This gives information on the immune status of HNSCC patients, which could be used for the development of novel (immuno)therapies [77]. Combination of single cell RNA sequencing (scRNA-seq) with multispectral imaging (MSI) will give the opportunity to characterize the spatial localization of immune cells and their cellular neighbourhood within the TME and therefore provide deeper insights into the immune cell repertoire of HNSCC, their transcriptional states and differentiation trajectories as well as into the cellular crosstalk in the TME with potential relevance to tumor progression [78,79,80].

In depth analyses of gene signatures as well as the composition of innate and adaptive immune cells in the TME of HPV^−^ and HPV^+^ HNSCC lesions highlighted significant differences in the repertoire and function of immune cells, which is reflected by an altered expression of inhibitory receptors and exhaustion markers [81] dependent on the HPV status of the tumor specimen, which might also have implications for their treatment with (immuno)therapies [81]. Higher immune cell infiltration in general, including T cells, together with an increased expression of activation markers like CD69, perforin and granzyme, as well as CD56^dim^ NK cells and Treg, B cells and an abundance of IFN-*γ* was found in HPV^+^ HNSCCs compared to HPV^-^ lesions [38,41,43,82,83]. In addition, the expression of different inhibitory molecules, such as PD-1, PD-L1, CTLA4 and TIM-3, was increased in HPV^+^, but not in HPV^-^ HNSCC [41,43].

The distinct immune cell repertoire in the TME might be associated with the status of the genomic HPV integration [84]. HPV antigens could then lead to the activation of primarily innate immune responses followed by adaptive immune responses mediated by T and B lymphocytes. Furthermore, single cell RNA sequencing analyses identified subpopulations of immune cells in particular of exhausted CD8^+^ T cells with different biological functions [85]. In the peripheral blood, HPV-specific CD4^+^ and CD8^+^ cells were more frequently observed in HPV^+^ than in HPV^-^ HNSCC patients [86].

## 5. Correlation of the Efficiency of (Immuno)Therapies with Immune Escape Mechanisms

It has been demonstrated that established therapies for HNSCC patients, such as radiotherapy, chemotherapy as well as mAbs directed against the EGF-R, affect the TME. Since CD8^+^ TILs exhibit a high frequency of PD1 and TIM-3 expression accompanied by high levels of granzymes and perforin and this is associated with the clinical outcome, a regulatory role for TIM-3 and PD1 was suggested in cetuximab-promoting cytolytic activities of CD8^+^ TILs. Furthermore, the increased frequency of PD-1^+^ and TIM-3^+^ CD8^+^ TILs was inversely correlated with the clinical outcome of cetuximab therapy [87].

Treatment of HNSCC patients with immunotherapies increases the overall survival (OS) of HNSCC patients suggesting that the immune system might be targeted by these drugs to achieve clinical benefits for these patients. Despite that 20–30% of HNSCC patients are being treated with mAbs directed against PD1 or its ligand PD-L1 had a better OS, the efficacy of iCPi is still limited [88]. Thus, there is an urgent need to improve the knowledge of the complex biology of this disease in particular of immune escape mechanisms, anti-tumoral immune responses and the composition of the TME in order to improve treatment efficacy [89]. These data might help to identify features associated with responsiveness to immunotherapy and might also lead to the design of novel treatment regimens using single or multi-agent immunotherapies alone or in combination with standard therapies. Some drugs targeting these distinct mechanisms are currently in the clinical development or are already approved for the treatment of HNSCC [1]. Next to targeting the PD1/PDL1 pathway, a number of novel immunotherapeutic targets are currently in preclinical studies and clinical phase I and II trials as single agent or in combination with other checkpoint molecules. These include drugs targeting LAG-3, TIM-3 and ICOS [90].

## 6. Impact of Immune Escape Mechanisms on HNSCC Patients’ Outcome

Both the direct immune escape mechanisms of tumor cells as well as the composition of the TME have an impact on the outcome and prognosis of the HNSCC patients. In this context, the frequency of both CD8^+^ and CD3^+^ T cells have been associated with an increased OS after chemoradiotherapy in HPV^+^ and HPV^-^ HNSCC [91,92,93]. In addition, HNSCC have been shown to possess a high degree of Treg infiltration [94,95,96], which correlated with a favorable OS [94,96]. This might reflect the downregulation of inflammation, which triggers the initiation of carcinogenesis [97]. When comparing HPV^+^ and HPV^-^ HNSCC, high levels of TILs were associated with improved survival in HPV^-^ HNSCC [98]. On the other hand, HPV^+^ tumors presented a less immunosuppressive tumor microenvironment with higher infiltration of CD8^+^ lymphocytes and presence of less Tregs when compared to HPV^−^ tumors [38]. In contrast, other studies speculate whether PD-L1 expression might be higher in HPV^+^ tumors [99]. Recently, tertiary lymphoid structures (TLS) with a high frequency of B cells were identified in HNSCC and found to be associated with an improved survival like in other tumor entities, whereas high frequencies of intra-tumoral B-lymphocytes rather indicated an adverse outcome [94,100]. Germinal center derived B cells were present during disease progression of HPV^+^ HNSCCs and with a reduced frequency in HPV^-^ HNSCC [101,102].

In a more recent multicenter study of patients with HNSCC after post-operative chemoradiotherapy, a high CD8^+^ TIL density in the tumor periphery, tumor stroma, and tumor cell area was predictive for improved OS [91]. In another study, only stromal TIL infiltration was associated with increased OS [103]. Concerning the clinical relevance of PD-L1 expression on the outcome of HNSCC patients, there exist controversial data. In a laryngeal HNSCC cohort, high PD-L1 expression assessed by Automated Quantitative protein Analysis (AQUA) positively correlated with disease outcome [104]. In a recent report by Yang and co-authors, PD-L1 was shown to correlate with improved progression-free survival (PFS), but not OS in patients with advanced HNSCC. As expected, patients with combined low frequency of TILs and high expression of PD-L1 were characterized by dismal survival [105]. Another retrospective analysis assessing the PD-L1 expression in a large cohort of patients demonstrated that high PD-L1 expression was the strongest negative predictor of patients’ outcome, independent of tumor stage and distant metastases [106]. In cancers of the oral cavity, increased PD-L1 expression has also been shown to correlate with a poor patients’ survival [107]. Furthermore, TIM-3, LAG-3, IDO and CTLA4 expression was negatively correlated with OS [108]. In addition, MDSC characterized by the markers CD11b^+^, CD14^+^, CD33^+^, HLA-DR^+^ were recently analyzed in the peripheral blood and TME of HNSCC patients and their presence was associated with an increased metastasis formation and disease recurrence [109]. In addition to levels of TILs, TAMs were also present in HNSCC lesions and their frequency correlated with a poor patients’ outcome [110].

## 7. Therapeutic Strategies to Overcome Immune Escape Mechanisms in HNSCC Patients

So far, the OS for patients with recurrent/metastatic (R/M) disease is only 10–13 months [111,112,113,114] when applying the current standard of care for locally recurrent disease (without surgical or radiation treatment options) and/or metastatic disease in the first-line setting with platinum-based doublet chemotherapy and cetuximab. Furthermore, second-line treatment options with cetuximab, methotrexate and taxane demonstrated a response rate between 10–13% and median PFS of 2–3 months without any obvious improvement in OS [115,116]. The data from Checkmate 141 and KEYNOTE-040 provided evidence for the use of single agent anti-PD-1 immunotherapy for the treatment of R/M HNSCC patients with disease progression after platinum-based chemotherapy. Furthermore, the KEYNOTE-048 study using anti-PD-1 therapy in patients with R/M HNSCC who had not received prior treatment with platinum-based chemotherapy also revealed a clinical benefit suggesting the use of pembrolizumab as an appropriate fist-line treatment for PD-L1^+^ HNSCC [117]. In 2016, the US Food and Drug Administration (FDA) granted the first immunotherapeutic approvals for the treatment of HNSCC patients with iCPi, namely the anti-PD-1 immune checkpoint inhibitors nivolumab and pembrolizumab, for the treatment of patients with HNSCC that are refractory to platinum-based regimens. The European Commission followed in 2017 with the approval of nivolumab and shortly thereafter with the approval of pembrolizumab monotherapy for the treatment of recurrent or metastatic HNSCC in adults, whose tumors express PD-L1 with a ≥ 50% tumor proportion score (TPS) and that have progressed after platinum-containing chemotherapy. In 2019, the FDA granted approval of pembrolizumab as first-line treatment for patients with metastatic or unresectable, recurrent HNSCC, in combination with platinum and fluorouracil and pembrolizumab as a single agent for patients with HNSCC, whose tumors express a PD-L1 combined positive score (CPS) ≥ 1. These approvals marked the first new therapies for these patients since 2006 as well as the first immunotherapeutic approvals in this disease entity.

However, to improve immunotherapy a much better understanding of emerging immunotherapies, including appropriate patients’ selection, influence of HPV infection, therapy sequence, response monitoring, adverse event management and biomarker testing, is required to guide improvements in care. In order to address these issues, the Society for Immunotherapy of Cancer (SITC) established the Cancer Immunotherapy Guideline—Head and Neck Cancer subcommittee to provide evidence-based recommendations on how to incorporate immunotherapies into practice for the treatment of patients with HNSCC [1].

The published data of clinical trials investigating the therapeutic response to checkpoint inhibitors clearly indicate that the HPV status does not predict the response to checkpoint inhibitors in HNSCC. Nevertheless, HPV^+^ and HPV^−^ HNSCC are pathogenetically different entities with different clinical outcomes. In this context, immunomodulatory therapies including specific vaccination are currently being tested in clinical studies and might improve the outcome especially of HPV^+^ tumors (among others NCT02163057, NCT02002182).

For both, HPV^+^ and HPV^−^ HNSCC, insights into the complex biology of this disease in particular of the anti-tumoral immune response and the TME is urgently required. This knowledge might help to identify features associated with non-responders and responders as well as the development of resistance to the immunotherapies. Based on this information, novel treatment regimens using single or multi-agent immunotherapies alone or in combination with standard therapies are being developed, which is summarized in Figure 1. Next to the PD1/PD-L1 pathways, some drugs targeting these distinct mechanisms are currently in preclinical studies, in clinical phase I and II trials as single agent or in combination with other checkpoint molecules or already approved for the treatment of HNSCCs. These include antibodies directed against LAG-3, TIM-3, TIGIT and ICOS [90,118,119,120]. In addition, a bifunctional fusion protein to block PD-L1 and TGF-β receptor has been applied in a phase I trial demonstrating a clinical activity [121].

Further approaches address the NK cell mediated antibody-dependent cellular cytotoxicity and the use of NK cell therapy, since the expression of HLA class I antigens is reduced up to 80% compared to the surrounding healthy tissues. [50,51,122]. NK cells might have the power to eliminate the HLA-I HNSCC cells via the missing self-recognition. However, HNSCCs express high levels of HLA-G and HLA-E [59,63], which inhibits both NK and CD8^+^ cells via ILT2/4 and/or NKG2A [123,124]. Therefore, mAbs targeting NKG2A, such as monalizumab, might be employed for the treatment of HNSCC by blocking the binding to HLA-E and/or HLA-G. Indeed, a phase II study of monalizumab combined with cetuximab showed an excellent activity in previously treated HNSCC patients including some previously exposed to cetuximab [63].

Since radiotherapy (RT), which is known to enhance immune responses, is a central option in the treatment of advanced HNSCC, the systemic immune modulatory influence of local radiation therapy in combination with immune checkpoint inhibition might improve the outcome of HNSCC patients [125]. Indeed, fractionated radiation together with immunotherapy might optimize tumor control by releasing dying cells that express mutated antigens together with adjuvant signals thereby eliciting an antigen-specific immune response [126,127,128,129,130]. Thus, it is important to emphasize that radiotherapy is a double-edged sword. While it exerts the tumoricidal effects and influences antigen presentation, expression of immune checkpoint receptors and recruitment of effector T cells, it has also the potential to create an immunosuppressive environment via the recruitment of MDSCs, TAMs and Tregs. Therefore, many trials are currently being performed to prove the efficacy of such combined therapies. It was demonstrated that patients treated with concurrent local radiotherapy and immune checkpoint inhibition had a longer OS when compared to those who received radiation either before or after starting immune checkpoint inhibition [131]. This might be a sign of an increased toxicity of combined therapies. Therefore, a number of clinical trials focus on safety issues by combining immune checkpoint inhibition with other therapeutic options, such as radiation and/or chemotherapies. For example, toxicity results from the GORTEC2015-01 (“PembroRad”) trial revealed that the frequency of grade 3 dermatitis, rash, and mucositis were significantly reduced in the pembrolizumab arm, while thyroid dysfunction was significantly increased compared to those treated with cetuximab [132]. Similar results were obtained by another small phase II trial [133], while other studies have evaluated the combination of immune checkpoint inhibition, radiotherapy and cisplatin. Taken together, there appears no major concerns of safety problems with this approach [134,135,136].

## 8. Conclusions

It is obvious that HNSCC are able to develop or use a multitude of mechanisms to escape T and/or NK cell-mediated immune surveillance. Thus, one single strategy might be often not sufficient to mount proper immune responses, since HNSCC evade immune recognition by different approaches, which explain the limited efficacy of immune therapies in this disease. Increased insights to the immune escape mechanisms will help in the design of novel as well as optimized (immuno)therapeutic strategies, as shown in Figure 1, which either have to be directed against tumor cells or revert the immune suppressive TME.

## Figures and Tables

**Figure 1 ijms-21-07032-f001:**
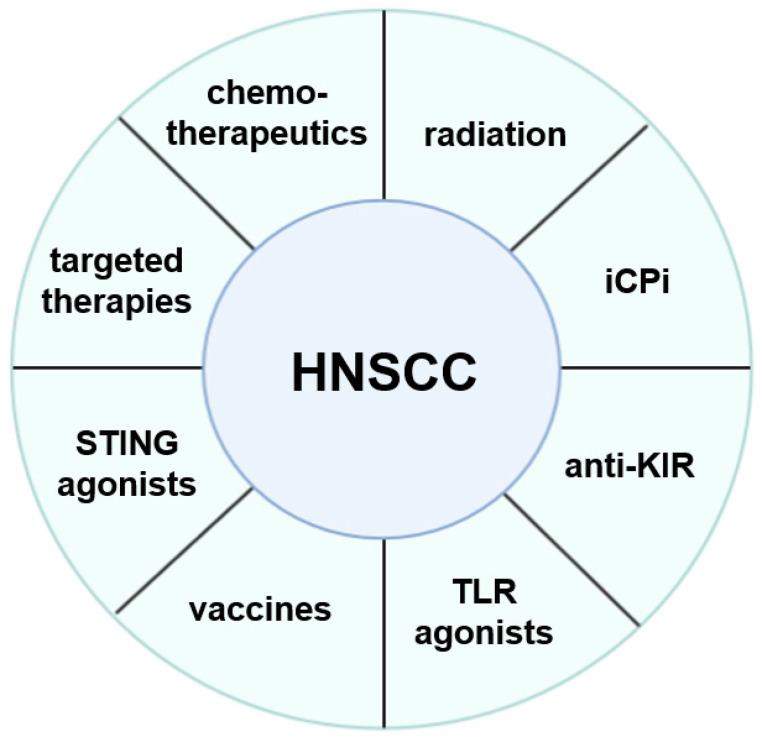
Different therapeutic options for the treatment of HNSCC.

**Table 1 ijms-21-07032-t001:** Distinct clinical and immunological features of HPV^−^ and HPV^+^ HNSCC.

Clinical Parameter	HPV^−^	HPV^+^
prevalence	95%	5%
risk factors	alcohol, tobacco	HPV
age	older	younger
localization	anywhere	mainly oropharynx
overall survival	worse	better
chemotherapy	-	better response
immunotherapy (iCPi)	response	increased response
tumor mutational burden	low	high
**Immunological Parameters**		
HLA-G	+	++
PD-L1	+	++
HLA class I loss	++	++
immune cell infiltration	+	++
immune suppression	+	++

+ moderate increase; ++ strong increase.

**Table 2 ijms-21-07032-t002:** Immune escape mechanisms of solid tumors.

Tumor	TME
MHC/HLA class I ↓	frequency and function CD8^+^ T cells ↓
APM ↓	frequency and function CD4^+^ T cells ↓
HLA-G/-E ↑	frequency Treg ↑
IFN pathway ↓	frequency and function NK cells ↓
PD-L1 ↑	frequency MDSC ↑
other checkpoint ligands ↑	frequency TAM ↑
adhesion molecules↓	
apoptosis-inducing genes ↑	
TGF-β, IL-10 ↑	frequency CAF ↑
metabolites ↑ arginase IDO NOS lactate	frequency and function monocytes ↓
acidic pH	
Hypoxia	

↓ downregulated, ↑ upregulated.

**Table 3 ijms-21-07032-t003:** Differences in the composition of the TME and its clinical relevance in HPV^+^ and HPV^-^ HNSCC lesions.

	HPV^−^		HPV^+^		Reference
immune cells/markers	frequency	clinical relevance	frequency	clinical relevance	[38]
CD4/CD8 TILs	low	improve when present	high	good prognosis	[39]
activation markers	low	bad	high	good prognosis	[40]
Treg	low		high	ratio CD8/Treg good outcome	[41]
NK cells (CD56dim)	high	improved prognosis when present	high	improved prognosis	[42]
B cells	low	n.a.	high	improved prognosis	[43]
M1/M2 ratio	low	worse outcome	high	good outcome	[44] [45]
MDSC	increased	increased metastasis	increased	increased metastasis	[46,47]
PD1/PD-L1	low		increased	increased metastasis	[48]

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
