# Peer review of "Immune Escape Mechanisms and Their Clinical Relevance in Head and Neck Squamous Cell Carcinoma"

_ijms, 2020, doi:10.3390/ijms21197032_

Round 1
Reviewer 1 Report
A very timely review with high scientific and clinical relevance about current knowledge on immune escape mechanisms of head and neck squamous cell carcinoma (HNSSC) is presented. The authors comprehensively summarize key general immune escape mechanisms of tumors and provide and discuss literature, which ones also occur in HNSCC. At the end, they suggest multimodal therapies for HNSCC to improve therapy response rates.
Nevertheless, some issues might additionally be taken into consideration:
- Particularly the final statement about different therapeutic options that might be combined to improve immunological anti-tumor immune responses (that are additionally listed in the only figure of this review) has to be outlined in more detail. For each mentioned treatment option a short rational should to be provided and discussed which combinations might be the most promising ones. E.g. immune modulatory effects of radiotherapy could be discussed at this point, as this is also mentioned lines 196-197, but not discussed or explained in more detail.
- The statement “The immune escape mechanisms of HNSCC cells and the HNSCC microenvironment are summarized in Table 1” should be placed after line 147 and not in line 116, since up to there only general tumor immune escape mechanisms and no specific ones or ones that were described for HNSCC were described.
- The authors have to shortly discuss whether the mentioned escape mechanism are specific for HNSCC or general ones for solid tumors.
- “EMT” should not be printed in bold.
Author Response
Reviewer #1:
1) Particularly the final statement about different therapeutic options that might be combined to improve immunological anti-tumor immune responses (that are additionally listed in the only figure of this review) has to be outlined in more detail. For each mentioned treatment option a short rational should to be provided and discussed which combinations might be the most promising ones. E.g. immune modulatory effects of radiotherapy could be discussed at this point, as this is also mentioned lines 196-197, but not discussed or explained in more detail.
- Thank you for this comment. As treatment options are manifold and there is an increased complexity regarding combination therapies, we have updated the section “Therapeutic strategies to overcome immune escape mechanisms in HNSCC patients” (beginning on page 12), providing a structured report to immune checkpoint inhibitors +/- other targeted therapies, followed by tumor vaccination, NK cell and radiotherapy. Furthermore, we provided current knowledge on differences between HPV+ HPV- HNSCC (see also new table 1 and 3).
2) The statement “The immune escape mechanisms of HNSCC cells and the HNSCC microenvironment are summarized in Table 1” should be placed after line 147 and not in line 116, since up to there only general tumor immune escape mechanisms and no specific ones or ones that were described for HNSCC were described.
- Thank you for recognizing this misplacement. We have moved the text segment to a more adequate place at the beginning of the introduction on page 4.
3) The authors have to shortly discuss whether the mentioned escape mechanism are specific for HNSCC or general ones for solid tumors.
- Thank you for that correction. Indeed, the immune escape mechanisms described can be generally attributed to malignant tumors and are not limited to HNSCC. We have added this statement in the text at the end of page 6 and provided a new table (table 2).
4.) “EMT” should not be printed in bold.
- Thank you for your notice. We have corrected the text at the end of page 6.
Reviewer 2 Report
This article is a clearly written and fairly comprehensive review of the literature on immune responses and immunomodulation in head and neck squamous cell carcinomas (HNSCC). It covers a large amount of work and will be a useful resource for clinicians and researchers.
A few updates would help to improve the utility of the review.
- HPV-positive (HPV+) and HPV-negative (HPV-) subtypes of HNSCC represent distinct disease subsets with some very different clinical and biological characteristics. HPV as a cause of HNSCC is introduced only very briefly (line 35). The review would benefit from some more description of the subtypes of HNSCC as they relate to HPV status (and perhaps to other characteristics such as anatomical site).
- Related to (1), it would be very helpful to include more description of how HPV+ and HPV- HSNCC differ with respect to the immune escape mechanisms discussed in the review. This could be achieved by a separate section of the manuscript that compares HPV+ and HPV- cancers, or by a statement in each section of what is and is not known about the influence of HPV status on the phenomenon being described. Some discussion of the work of Dohun Pyeon’s group might be beneficial to include in the manuscript.
- Related to the comments above, line 188 refers to differences in the immune cell repertoire with HPV status, but the nature of the differences is not specified.
- On line 84 the text “X, Y, and Z” appears to be a placeholder for the specific gene/protein names.
Author Response
Reviewer #2:
1) HPV-positive (HPV+) and HPV-negative (HPV-) subtypes of HNSCC represent distinct disease subsets with some very different clinical and biological characteristics. HPV as a cause of HNSCC is introduced only very briefly (line 35). The review would benefit from some more description of the subtypes of HNSCC as they relate to HPV status (and perhaps to other characteristics such as anatomical site).
- Thank you for your comments. As HPV+ HPV- HNSCC differ strongly in their prevalence, molecular signature, localization, prognosis, therapy response and immune cell infiltration, we have highlighting and expanded these differences in two sections of our manuscript. General clinical and immunological characteristics are provided in the introduction on page 4 with tables 1 and 3. Differences in treatment options are discussed on pages 9-13 along with table 1.
2) Related to (1), it would be very helpful to include more description of how HPV+ and HPV- HSNCC differ with respect to the immune escape mechanisms discussed in the review. This could be achieved by a separate section of the manuscript that compares HPV+ and HPV- cancers, or by a statement in each section of what is and is not known about the influence of HPV status on the phenomenon being described. Some discussion of the work of Dohun Pyeon’s group might be beneficial to include in the manuscript.
- Thanks for your comments. We have addressed this issues in the section “Immune escape mechanisms of tumors” and provided a summary of the current knowledge, which can be found in table 2. In addition, we cited the most important work of the group of Pyeon (page 7, reference 47).
3) Related to the comments above, line 188 refers to differences in the immune cell repertoire with HPV status, but the nature of the differences is not specified.
- Based on this comment we have provided additional information to the relation of immune cell repertoire with HPV status as depicted on pages 9-10 at the end of the section “Tumor microenvironment and HNSCC”.
4) On line 84 the text “X, Y, and Z” appears to be a placeholder for the specific gene/protein names.
- We would like to state that the constitutive subunits of the multicatalytic proteasome complex are called MB1 (X), Delta (Y) and ß2 (Z) and did not represent placeholder.
This manuscript is a resubmission of an earlier submission. The following is a list of the peer review reports and author responses from that submission.